# Influence of Biologically Oriented Preparation Technique on Peri-Implant Tissues; Prospective Randomized Clinical Trial with Three-Year Follow-Up. Part I: Hard Tissues

**DOI:** 10.3390/jcm8122183

**Published:** 2019-12-11

**Authors:** Rubén Agustín-Panadero, Naia Bustamante-Hernández, María Fernanda Solá-Ruíz, Álvaro Zubizarreta-Macho, Antonio Fons-Font, Lucía Fernández-Estevan

**Affiliations:** 1Department of Dental Medicine, Faculty of Medicine and Dentistry, University of Valencia, 46010 Valencia, Spain; rubenagustinpanadero@gmail.com (R.A.-P.); m.fernanda.sola@uv.es (M.F.S.-R.); antonio.fons@uv.es (A.F.-F.); lucia.fernadez-estevan@uv.es (L.F.-E.); 2Department of Implant Surgery, Faculty of Health Sciences, Alfonso X el Sabio University, 28691 Madrid, Spain; azubimac@hotmail.com

**Keywords:** implant-supported prosthesis, bone loss, hard tissue, screwed, cemented BOPT

## Abstract

Purpose: The objective of this prospective randomized clinical trial (RCT) was to analyze and compare the clinical behavior of three types of prosthesis supported by single implants in the posterior region after three years functional loading. Materials and Methods: Seventy-five implants were divided into three groups according to the type of prosthetic restoration: screw-retained crown (Group GS); cemented crown without finishing line (biologically oriented preparation technique) (Group GBOPT); and conventional cemented crown with finishing line (Group GCC). The clinical behavior of each restoration type was analyzed after 3 years functional loading by analyzing radiographic peri-implant bone loss. Results: GBOPT obtained the least bone loss (*p* < 0.01) in comparison with GS and GCC. Conclusions: Bone loss around implants is related to the type of prosthetic restoration it supports, whereby cemented BOPT crowns present less bone loss.

## 1. Introduction

Rehabilitation with dental implants has become a frequently used procedure enjoying high success rates of between 90% and 95% after ten yeas [1,2,3]. Among the many factors that influence implant survival are the patient’s health status, the characteristics of the host site, and the surgical technique employed [4,5,6,7,8]. With regard to risk factors derived from the patient’s health, the most important are deficient plaque control, unmanaged periodontal disease, smoking, particular medications that affect bone remodeling, and systemic diseases such as diabetes or generalized osteoporosis [9,10,11]. As for host site characteristics, dimensions and bone tissue conditions, implant position, and the surgical procedure used all influence the implant’s biological behavior [12,13]. Implants and the prosthetic restorations they support vary in terms of whether implants are placed at bone level or tissue level, the macroscopic morphology of the implant body, the implant’s surface treatment, and differences in the type of implant-supported restorations. Restorations may be screw-retained or cemented onto the implant abutment; they can also be screwed and cemented, presenting the characteristics of both types of retention using an intermediate titanium abutment supporting a monolithic crown cemented to it with a perforation in the occlusal face. The various options have obtained similar results in diverse clinical trials [14], although cemented restorations have been observed to suffer more biological complications such as peri-implant inflammation, due to extravasation of the cement [15,16,17,18]. Screw-retained restorations are related to higher numbers of mechanical complications such as loosening and fracture of the fixation screw [14,18,19].

In recent years, an alternative to the conventional cementation technique has developed that eliminates the finishing line, replacing it with a telescopic crown with ovoid emergence located 1 mm subgingivally, inside the peri-implant sulcus (Figure 1).

This type of restoration without finishing line is also used for tooth-supported full coverage crowns and is known as biologically oriented preparation technique (BOPT). The design improves peri-implant tissue behavior as it eliminates the gap between the restoration and the finishing line at the end of the transepithelial abutment [20,21]. The BOPT approach aims to model the peri-implant soft tissues through the use of conical abutments, leaving the apical portion of the abutment free of coverage by the prosthetic restoration for at least 2 mm in order to stabilize the adjacent connective tissue [22,23,24]. By reducing the thickness of the apical part of the abutment adjacent to the implant’s prosthetic platform (this being convergent rather than divergent), it allows for greater peri-implant soft tissue thickness, avoiding its subsequent recession and/or thinning of the mucosa, which aids medium-to long-term stability [25] (Figure 2A,B).

The objective of this prospective randomized clinical trial (RCT) was to analyze peri-implant bone tissue behavior around three types of single implant-supported prosthetic crown (screw-retained, cemented BOPT, and cemented with finishing line) after three years functional loading. In addition, the bone loss of screw-retained crowns, crowns cemented slightly subgingivally without finishing line (BOPT), as well as crowns cemented conventionally with finishing line placed slightly subgingivally were compared. A comparative analysis was also performed between screwed and cemented prostheses (without distinguishing between BOPT and conventional cementation). The mean bone loss between the two cementation techniques (BOPT/conventional) was also analyzed. Differences between the maxilla and mandible were analyzed for each type of prosthesis. The hypothesies tested were that there was (h1) higher marginal bone loss in cemented crowns compared with the screw-retained restorations, (h2) there was similar bone loss between cemented crowns (BOPT/conventional), and (h3) there was more peri-implant bone loss in the implants placed in the maxilla than in those in the mandible.

## 2. Materials and Methods

This prospective RCT was conducted at the dental clinic of the Prosthodontic and Occlusion Unit, Faculty of Dentistry and Medicine, University of Valencia (Carrer Gasco Oliag n°1, 46010, Valencia, Spain). The trial was approved by the University of Valencia Ethics Committee for Research Involving Humans (registration number 111508667100076). All participants were provided with full information about the trial, its objectives, and the procedures involved, and all gave their informed consent to take part before the trial began. The study sample consisted of 75 implant-supported crowns. Patients were treated between January 9th and December 16th, 2015. All participants presented good general health, without any contraindications to oral surgery, and a single missing tooth in the molar or premolar region.

### 2.1. Inclusion/Exclusion Criteria

Inclusion criteria were as follows: patients with single edentulism in the posterior region (molar and premolar regions); patients older than 18 years; patients willing to attend follow-up visits scheduled during the trial (at 3 and 6 months, and 1, 2, and 3 years after surgery); patients in good general health, without any factors that would affect implant stability, such as joint/bone pathologies or patients presenting keratinized mucosa in the vestibular aspect of the edentulous area of at least 2 mm.

Exclusion criteria were: patients with partial or complete edentulism (more than one tooth missing); patients requiring guided bone regeneration in addition to implant placement; patients with less than 7 mm available prosthetic height; pregnant or lactating women; patients with systemic or local diseases or other contraindications to implant placement (previous chemotherapy or radiation to the head and/or neck, progressive active periodontitis, and/or immunosuppression), or patients with implants adjacent to the edentulous area (all patients presented natural teeth adjacent to the edentulous space).

### 2.2. Surgical Procedure

All patients were treated by the same surgeon (A-P, R). The implants used in the trial were Vega Klockner implants (Escaldes-Engordany, Andorra) with internal hexagonal connection associated with conical geometry. These implants are fabricated from grade IV titanium with two-stage surface treatment (alumina sandblasting and thermochemical treatment). All implants were placed following the same surgical protocol: implants were placed at bone crest level in nongrafted mature bone using full-thickness surgical flap without releasing, making a supracrestal intrasulcular incision from the adjacent teeth (Figure 3).

After surgery, 1 g Amoxicillin was administered (GlaxoSmithKline, Madrid, Spain), or Clindamycin 300 mg in case of allergy, one every 8 h for 7 days; and 600 mg ibuprofen (Bexistar, Laboratorio Bacino, Barcelona, Spain) as postoperative medication. As coadjuvant treatment, 0.12% Chlorhexidine mouthwash was prescribed (GUM, John O. Butler/Sunstar, Chicago, IL, USA) three times per day.

After surgery, patients returned for check-ups one week, one month, and three months later, when prosthetic loading was performed.

### 2.3. Sample Randomization

After surgery, the types of restoration prosthetics were distributed randomly forming three groups: Group GS, implants restored with CAD/CAM crowns (Cr–Co with feldspathic ceramic coating) screwed directly onto the implants; Group GBOPT, implants restored with crowns cemented slightly subgingivally (0.5 mm) onto grade IV titanium abutments without finishing line (BOPT); Group GCC, implants restored with crowns cemented conventionally onto grade IV titanium abutments with 1 mm wide chamfered finishing line placed slightly subgingivally (0.5 mm).

### 2.4. Sample Size Calculation

A previous study of the sample size (power analysis) was carried out, concluding that a minimum of 75 implants (25 per group) were needed to achieve a power of 80% in order to detect as significant an effect size f = 0.35 (half-large) in the average MBL (mean bone loss) of groups, for 95% confidence. Standard deviation 0.45 mm was assumed, so that effect size is compatible with average MBL in groups 0.25 mm, 0.45 mm, and 0.65 mm.

The three restoration types (total = 75) were allotted to patients using online randomization software: www.alazar.info.

### 2.5. Prosthetic Crown Fabrication

Digital impressions used for fabricating the crowns were taken with an intraoral scanner (True Definition, 3M ESPE, Two Harbors, MN, USA).

All the metal–ceramic crowns were milled from Cr–Co (Archimedes, Klockner S.A. Barcelona, Spain) and coated with feldspathic ceramic (IPS d-Sign, Ivoclar Vivadent, Schaan, Liechtenstein) by means of CAD/CAM design software (EXOCAD DENTAL CAD, Exocad America, Inc., STI Holdings, Woburn, MA, USA) (Figure 4).

Group GS crowns were screwed onto implants using a titanium screw applying torque of 30 N/cm. For cemented crowns, both with and without finishing line, grade IV titanium abutments were screwed onto the implants applying torque of 30 N/cm, fixing the crowns in place with provisional cement (Premier Implant Cement, Premier Dental, Plymouth Meeting, PA, USA). Excess cement was cleaned following established protocol: interproximal dental floss is placed before cement setting, cleaning is done with silk towards the prosthetic platform of the implant, taking as advantage that the cement is radiopaque and the excess can be seen in a radiographic control, then the excesses are removed with a fine instrument.

Changes in peri-implant bone were evaluated from standardized parallelized periapical radiographs captured using a Rinn positioner (Rinn, Dentsply Sirona Pt Ltd., St Leonards NSW, Australia), observing bone levels at the moment of prosthetic loading (Tinitial) and again 3 years after loading (Tfinal). Radiographs were processed and stored using an intraoral digital system (RVG Carestream, Carestream Health, Inc, Rochester, NY, USA).

Measurements were taken by two independent observers (B-H, N) (F-E, L), and mean values were calculated to avoid different interpretations of the radiographs. After prosthetic loading, patients returned for follow-up check-ups after 1 month, 6 months, and 1, 2, and 3 years.

### 2.6. Radiological Marginal Bone Level Evaluation

Peri-implant bone changes were evaluated by means of standardized parallelized periapical radiographs using an X-ray positioner (Rinn, Dentsply Sirona Pt Ltd., St Leonards NSW, Australia). To avoid radiographic distortion, the width of the prosthetic platform of each implant (in millimeters) was used as a reference value. This width was provided by the manufacturer, printed on the product label. As the implants were placed at crestal bone level, the straight line between the mesial and distal sides of the implant was considered as height 0 (corresponding to the implant platform). To determine bone loss, a line was traced perpendicularly from points on the mesial and distal aspects of each implant to the most coronal bone level. Measurements were taken to evaluate bone changes at the moment of prosthetic loading (initial) and 3 years later (final). In this way, bone loss was measured on both mesial and distal sides of the implant on each radiograph. The measurements were taken using three-dimensional (3D) modeling software (Rhinoceros^®^, Robert McNeel & Associates, WA, USA) (Figure 5A,B).

Intra-examiner calibration was analyzed before evaluating the entire implant sample by reassessing bone loss at a total of 30 randomly selected sites (using the random function of Microsoft Excel 2010), taking duplicate measurements on different days. An intraclass correlation coefficient of 0.898 was obtained, showing high concordance between the two sets of data. According to Dahlberg’s *d* value, 0.049 mm error was estimated for the measurement method.

### 2.7. Statistical Analysis

Statistical analysis began by calculating descriptive statistics for the bone loss variable (mean, standard deviation, range, and median). Nonparametric tests were applied. The Chi-square test of association was used to analyze the rate of incidence of bone loss for each type of prosthesis, and the fisher exact test for comparing pairs of groups. The Kruskal–Wallis test was used to measure the homogeneity of the bone loss variable by group. The Mann–Whitney test was applied to compare pairs of independent groups. Data provided a power of 70.5% to detect a medium–large effect size (*d* = 0.65) for differences in bone loss between groups, assuming 95% confidence. The significance level was set at 5% (*p* < 0.05%).

## 3. Results

### 3.1. Study Population

This prospective RCT included 75 implants, placed in 75 patients (49 women and 26 men, with a mean age of 42.7 ± 10.6 years) (Table 1).

Patients/implants were divided into three groups (*n* = 25) according to the type of prosthetic restoration used. Radiographic evaluation of peri-implant bone loss was carried out for each group after 3 years functional loading. It should be noted that seven patients were lost to the trial due to their failure to attend follow-up visits, leaving a sample of 68 implants in 68 patients.

Among the total implant sample, 42.6% were placed in the maxilla and 57.4% in the mandible; 55.9% of the implants were placed in molar regions, and 44.1% in premolar regions.

### 3.2. Bone Levels

Radiographic peri-implant bone loss showed differences among the three types of restoration used: Group GS (screw-retained) = 0.35 ± 0.37 mm; Group GBOPT (cemented BOPT) = 0.10 ± 0.28 mm; and Group GCC (conventional cementation with finishing line) = 0.67 ± 0.62 mm. Median bone loss values were 0.36 mm, 0.00 mm, and 0.63mm, respectively (Table 2).

Bone loss (mm) for the three types of prosthesis and two cementation techniques (BOPT/conventional) was analyzed. No statistically significant differences were found between GS and GCC (*p* = 0.059, Kruskal–Wallis and Mann–Whitney tests), although less bone loss was observed in Group GS. However, significant differences were found between Groups GS and GBOPT (*p* = 0.013, Kruskal–Wallis and Mann–Whitney tests) whereby the GBOPT Group underwent less bone loss. In the same way, differences were also found between the GBOPT and GCC Groups (*p* < 0.001, Kruskal–Wallis and Mann–Whitney tests), Group GBOPT presenting less bone loss. The rate of incidence of bone loss for the three prosthetic types and two cementation techniques (BOPT/conventional) showed differences between Group GS and Group GBOPT (*p* = 0.011, Chi^2^) and between Group GBOPT and Group GCC (*p* < 0.001, Chi^2^) but not between Group GS and Group GCC (*p* = 0.058, Chi^2^), although Group GS presented less bone loss than GCC Group (Figure 6).

The influence of implant position (maxillary/mandibular) on bone loss in relation to the type of prosthesis used and in relation to the two cementation techniques (BOPT/conventional) was analyzed. For implants placed in the maxilla, no statistically significant differences were found among the three groups: Groups GBOPT and GCC (*p* = 0.135, Kruskal–Wallis and Mann–Whitney tests); Groups GS and GCC (*p* = 0.758, Kruskal–Wallis and Mann–Whitney test); or Groups GS and GBOPT (*p* = 0.235, Kruskal–Wallis and Mann–Whitney tests). But for implants placed in the mandible, statistically significant differences were found between Groups GS and GCC (*p* = 0.049, Kruskal–Wallis and Mann–Whitney tests) and between Groups GBOPT and GCC (*p* = 0.002, Kruskal–Wallis and Mann–Whitney tests); differences between Groups GS and GBOPT did not reach significance (*p* = 0.153, Kruskal–Wallis and Mann–Whitney tests), although Group GBOPT (cemented BOPT) presented less bone loss. The rate of incidence of bone loss for the three prosthetic types and for cementation techniques (BOPT/conventional) in the maxilla did not show statistically significant differences between any of the three groups, neither between the Group GS (screwed) and Group GCC (conventional cementation) (*p* = 1.000, Chi^2^ Fisher tests), between Group GS and Group GBOPT (cemented BOPT) (*p* = 0.178, Chi^2^ and Fisher tests), not between Group GBOPT and Group GCC (*p* = 0.122, Chi^2^ and Fisher tests). Regarding the rate of incidence of bone loss in the mandible, statistically significant differences were found between Group GBOPT and Group GCC (*p* = 0.002, Chi^2^ and Fisher tests), with less bone loss in Group GBOPT (cemented BOPT). But no differences were found between Group GS and Group GCC (*p* = 0.083, Chi^2^ and Fisher tests), or between Group GS and Group GBOPT (*p* = 0.193 Chi^2^ and Fisher tests) (Figure 7).

Differences between the maxilla and mandible were also analyzed for each type of prosthesis. Bone loss (mm) in relation to screw fixation (GS) in the maxilla or mandible did not obtain significant differences (*p* = 0.559, Mann–Whitney test). Nor were there any differences in the rate of incidence of bone loss between maxilla and mandible (*p* = 0.383, Fisher test). The BOPT cemented prostheses (Group GBOPT) did not show significant differences in bone loss between maxilla and mandible either (*p* = 0.750, Mann–Whitney test), or in the rate of incidence of bone loss (*p* = 0.618, Fisher test). For the conventionally cemented prosthesis (Group GCC), no significant differences were found between maxilla and mandible for either bone loss (*p* = 0.142, Mann–Whitney test) or rate of incidence of bone loss (*p* = 0.077, Fisher test), although the rate of incidence was higher in the mandible.

A comparative analysis was also performed between screwed and cemented prostheses (without distinguishing between BOPT and conventional cementation). In this way, two groups were compared, one with a sample of 25 screwed prostheses, and the other with a sample of 43 cemented prostheses (21 BOPT and 22 conventionally cemented). Comparing bone loss (mm) between screwed and cemented crowns, the difference was not statistically significant (*p* = 0.872, Mann–Whitney test). Nor were differences in the rates of incidence of bone loss found between these groups (*p* = 0.700, Chi^2^). No differences were found between the groups (screwed/cemented) in relation to the arch in which they were placed (maxillary/mandibular) with regard to bone loss in the maxilla (*p* = 0.501, Mann–Whitney test) or the rate of incidence of bone loss (*p* = 0.449, Chi^2^), or with regard to bone loss in the mandible (*p* = 0.471, Mann–Whitney test) or the rate of incidence of bone loss (*p* = 0.645, Chi^2^) (Figure 8).

Comparing bone loss undergone by screwed prostheses in relation to their placement in the maxilla or mandible, no significant differences were found between maxilla and mandible (*p* = 0.559, Mann–Whitney test), nor were there differences in the rate of incidence (*p* = 0.383, Chi^2^). For cemented prostheses, statistically significant differences in bone loss were found (*p* = 0.011, Mann–Whitney test), whereby Group GBOPT obtained less bone loss and a lower rate of incidence in the maxilla (*p* = 0.010, Chi^2^), so a higher number of affected implants and higher bone loss in the mandible.

### 3.3. Regression Analysis to Explore Sex, Age, and Position Factors

A regression model to explain the MBL based on gender and age variables, as well as the position of the implant was made. The intention was to assess its direct influence on MBL and adjust the main relationship (the effect of the type of restoration) by this series of variables that may exert some kind of confusion in the event that the groups of restorations are not homogeneous and there is some kind of bias. The model-dependent variable was the MBL. Independent variables were Group (screwed/cemented conventional/cemented BOPT), Location (maxilla/mandible), Position (premolar/molar), Sex (male/female), and Age. The analyses result in fairly homogeneous distributions in the three groups of restorations. Note that the overall average age fits perfectly with the data registered: 43.4 ± 10.4 years. The interpretation of the model and its coefficients is represented as follows (Table 3):

An important factor in explaining the MBL is the position. In the molars, there is an MBL significantly higher than that of the premolars (beta = 0.273; *p* = 0.038). The maxilla/mandible position or demographic profile has no relevant influence on the MBL. Regarding the quality of fit of the model, the coefficient of determination, R2, was 31% and the adjusted R2 was 24.2%, which means that in terms of significant relationships beyond the findings, the equation is not good for making accurate MBL predictions. Despite the lack of normalcy of the dependent variable, especially in some groups with the cemented BOPT, the model residues fit reasonably well to the normal pattern, even with some tendency to positive asymmetry.

## 4. Discussion

This 3 year RCT revealed differences in amounts of bone loss in relation to the different types of prosthetic restoration analyzed.

In response to the first of the objectives of the present investigation, which was the analysis comparing bone loss (mm) by restoration type; it was observed Group GBOPT (cemented crown without finishing line, or BOPT) presented smaller changes in marginal bone level. Significant differences were found between groups GS and GBOPT (*p* = 0.013, Kruskal–Wallis and Mann–Whitney tests). In the same way, differences were also found between the GBOPT and GCC Groups (*p* < 0.001, Kruskal–Wallis and Mann–Whitney tests), presenting GBOPT less bone loss in both cases. The rate of incidence of bone loss for the three prosthetic types and two cementation techniques (BOPT/conventional) showed differences between Group GS and Group GBOPT (*p* = 0.011, Chi^2^) and between Group GBOPT and Group GCC (*p* < 0.001, Chi^2^), showing again less incidence on GBOPT. According to these results, the hypothesis (h1) that higher marginal bone loss in cemented crowns was obtained compared with the screw-retained restorations was rejected.

A comparative analysis was also performed between screwed and cemented prostheses (without distinguishing between BOPT and conventional cementation). It was observed that for cemented prostheses, statistically significant differences in bone loss were found (*p* = 0.011, Mann–Whitney test), whereby Group GBOPT obtained less bone loss and a lower rate of incidence in the maxilla (*p* = 0.010, Chi^2^), so a higher number of affected implants and greater bone loss in the mandible. A hypothesis (h2) about similar bone loss between cemented crowns (BOPT/conventional) was also rejected.

Another variable analyzed in the present RCT was whether the implants location (maxillary or mandibular) influenced the amount of bone loss. For implants placed in the mandible, statistically significant differences were found between Groups GS and GCC, presenting GS less bone loss (*p* = 0.049, Kruskal–Wallis and Mann–Whitney tests) and between Groups GBOPT and GCC, obtaining less bone loss in GBOPT (*p* = 0.002, Kruskal–Wallis and Mann–Whitney tests). Regarding the rate of incidence of bone loss in the mandible, statistically significant differences were found between Group GBOPT and Group GCC (*p* = 0.002, Chi^2^ and Fisher tests), with less bone loss in Group GBOPT (cemented BOPT). The hypothesis (h3) about higher peri-implant bone loss in the implants placed in the maxilla than in those in the mandible was rejected, according to the results the mean bone loss was higher in mandible.

This prospective RCT evaluated radiographic peri-implant bone loss after 3 years functional loading around three types of implant-supported prosthetic restoration, one of the most widely used parameters for evaluating implant success is the quantification of vertical bone loss around implants. The main limitation of this work was that the evaluation of 3D structures was measured in two-dimensional (2D) radiographs. Although, it is important to highlight that it is a method widely used that allows sufficient information for the follow-up of a single implant and it also beams less radiation for the patient. Other limitations of the present trial were, firstly, it would be preferable to recruit a larger sample; secondly, it could be interesting to analyze the influence of the height of the abutment on peri-implant bone loss; and thirdly, it could be also interesting to analyze the behavior of the same restoration types in the anterior region. It would also be useful to prolong the follow-up period beyond 3 years.

The lack of standardization could be another drawback. Since the different radiographs may have been performed from a different distance, modifying the size of the implant and surrounding bone to be analyzed, the images were scaled in this study using as a reference vale the width of the prosthetic platform of each implant (in millimeters). In order to assess the bone loss produced, bone loss was evaluated as used previously by numerous researchers such as Sanz-Martín et al. [25]; Callan et al. [26]; Patri et al. [27]; Peñarrocha-Diago et al. [28]; Agustín-Panadero et al. [3]; Canullo et al. [24], and Marconcini et al. [29]. Bone loss measurement used Rhinoceros software, as in an earlier study by Agustín-Panadero et al. [3]. The software processes the images captured after calibrating the radiographs as described by Piao et al. [30]; Peñarrocha-Diago et al. [28]; Spinato et al. [31]; and Agustín-Panadero et al. [32], who used various software tools with the same objective as the present trial.

The lowest bone loss values that are related in all cases to GBOPT could be due to the supracrestal position of the implant–abutment junction, which limits bacterial access, minimizing the inflammatory response from bacterial contamination, resulting in a reduction in peri-implant crestal bone loss [24]. These results could be related to those obtained by Agustín-Panadero et al. [1], in which the smallest changes in marginal bone were seen in a group with an internal hexagon connection, a tissue-level (supracrestal) implant with a 2.8 mm convergent transmucosal collar, obtaining a mean loss of 0.24 mm; SD (standard deviation): 0.22 mm. Similar MBL was obtained in this investigation, specifically in Group GBOPT, with a mean loss of 0.10 ± 0.28 mm.

Various authors have already demonstrated differences in bone loss around cemented and screw-retained implant-supported prostheses [21,33]. In the split-mouth trial by Nissan et al., which investigated implant-supported partial restorations, it was found that marginal bone loss was significantly (*p* < 0.001) higher with screw-retained (1.4 ± 0.6 mm) than cemented restorations (0.69 ± 0.5 mm), this result could be because in partial restorations, the gap that occurs could be supplied by cement, avoiding the bone loss that could occur due to this gap [33]. However, other trials that have compared bone loss between screw-retained and cemented restorations have not observed statistically significant differences in peri-implant bone loss between the two types [20,34,35].

Although some research has investigated implant-supported BOPT type prosthetics [22,23,24,29], none has compared BOPT with a control group or with conventional restorations, whether screw-retained or cemented over a horizontal finishing line. Moreover, no trial has compared the three prosthetic types analyzed in the present work, making any comparison of the present results with existing literature impossible.

Various studies have looked into differences in bone loss around implants placed in the maxilla and the mandible, but, like the present work, they have not observed significant differences; nor have differences been found between distal and mesial bone loss [36,37,38]. However, a trial by Peñarrocha et al. did identify statistically significant differences in peri-implant bone loss between implants placed in the maxilla and mandible [39]. In another study [32], bone loss was higher in mandible, mean bone loss was greater for implants with divergent transmucosal morphology (0.72 mm) than for those with convergent (0.19 mm), with a statistically significant difference (*p* = 0.028), according to the results obtained in this study.

Various works in the literature have obtained less bone loss results with convergent BOPT abutments than with other types of abutment [3,24,40]. This was affirmed by Canullo et al., who obtained a bone loss of 0.071 ± 0.11 mm [24], but a control group was not present, the location of implant was anterior area, and a patient selection was not randomized. As well as, Manconcini et al., who obtained a mean marginal bone level of 1.39 ± 0.91 mm at the moment of the prosthetic-transfer connection for definitive impression-taking. One year after loading, the mean marginal bone level reached was 1.16 ± 0.911 mm, with an average overall change of 0.18 ± 0.72 mm [29]. Even though the results were similar, the follow-up period was only a year and it did not compare it with control groups. The results obtained in this investigation were similar with a mean loss of 0.10 ± 0.28 mm. The results obtained could be related to the type of transepithelial abutment used. Convergent abutments increase the space available for connective tissue, favoring increased tissue thickness [41,42], resulting in maintenance of the adjacent hard tissues.

This prospective RCT could help on clinicians election for the restoration of an implant-supported unit posterior region because, in terms of bone loss, it would be recommended to place BOPT cemented restoration applying a rigorous protocol for cleaning cement excesses, as well as analyzing and studying the space left for the connective tissue under the restoration in contact with the prosthetic abutment. For clinicians who do not want to increase the prosthodontic difficulty of placing a cemented BOPT crown due to cement extravasation, based on the results obtained they could be advised the use of screw-retained crowns directly to the implant, taking into account that the crown’s prosthetic platform could be altered in the cooking processes of the ceramic coating. Resulting in the possibility of a higher rate of screw loosening or mechanical complications at medium–long term.

Further RCTs of BOPT are needed to analyze the clinical behavior of implant-supported restorations, as well as in vitro microbiology studies to analyze the bacterial proliferation capacity between restoration and implant. Additional studies are also necessary to determine the role played by different types of prosthesis in the incidence of complications and peri-implant pathologies.

## 5. Conclusions

According to the results of the present trial, it may be concluded that:

(1) Cemented BOPT-type prostheses may suffer less peri-implant bone loss after a 3-year follow-up, while conventionally cemented prostheses with finishing line obtain higher bone loss after 3 years.

(2) Comparing screw-retained with cemented prostheses (without distinguishing between BOPT and conventional), no differences are found in the incidence of bone loss.

(3) No statistical differences in peri-implant bone loss exist in relation to placement in the maxilla or mandible.

## Figures and Tables

**Figure 1 jcm-08-02183-f001:**
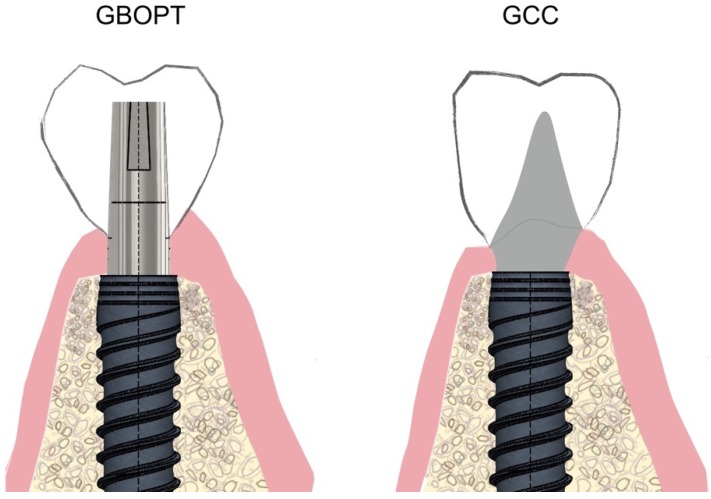
Schematic drawing of biologically oriented preparation technique (BOPT, Group GBOPT) compared with conventional cemented crowns (GCC) on implants.

**Figure 2 jcm-08-02183-f002:**
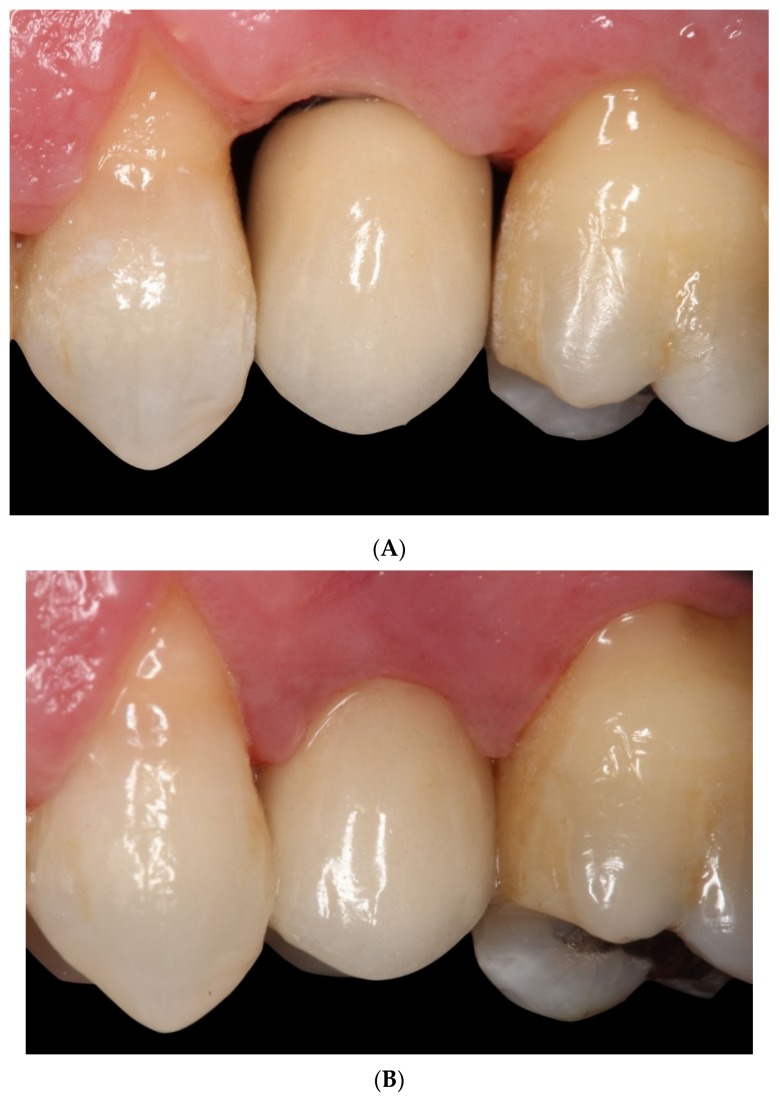
(**A**) Crown cemented without finishing line (BOPT) at the moment of prosthetic loading. (**B**) Crown cemented without finishing line (BOPT) 6 months later.

**Figure 3 jcm-08-02183-f003:**
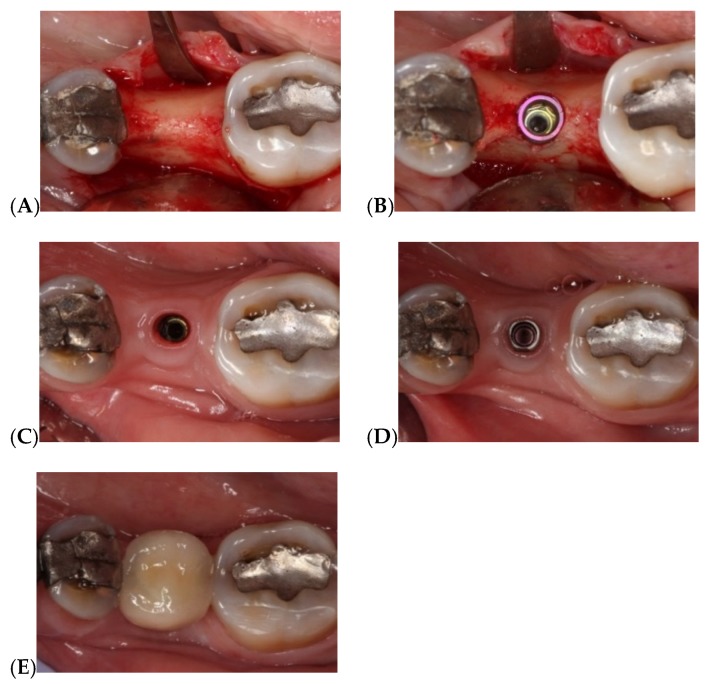
Implant-supported single restoration in the posterior region: (**A**) Exposing bone tissue after raising supracrestal intrasulcular full-thickness flap; (**B**) Crestal placement of Vega Klockner implant at bone level; (**C**) Keratinized mucosa after 3 months osteointegration; (**D**) Placement of titanium BOPT abutment (without finishing line) screwed to the implant’s prosthetic platform; (**E**) Restoration cemented onto abutment with Premier temporary implant cement.

**Figure 4 jcm-08-02183-f004:**
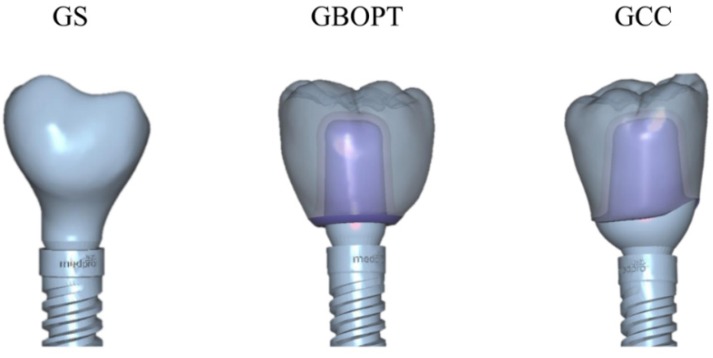
Three different implant-supported crown designs: GS, screw-retained crown; GBOPT, cemented without finishing line; GCC, conventional cementation with finishing line.

**Figure 5 jcm-08-02183-f005:**
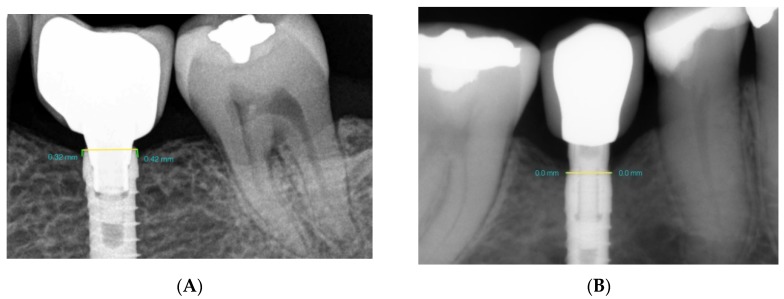
(**A**) GS and (**B**) GBOPT. Radiographic measurement of bone loss (mm).

**Figure 6 jcm-08-02183-f006:**
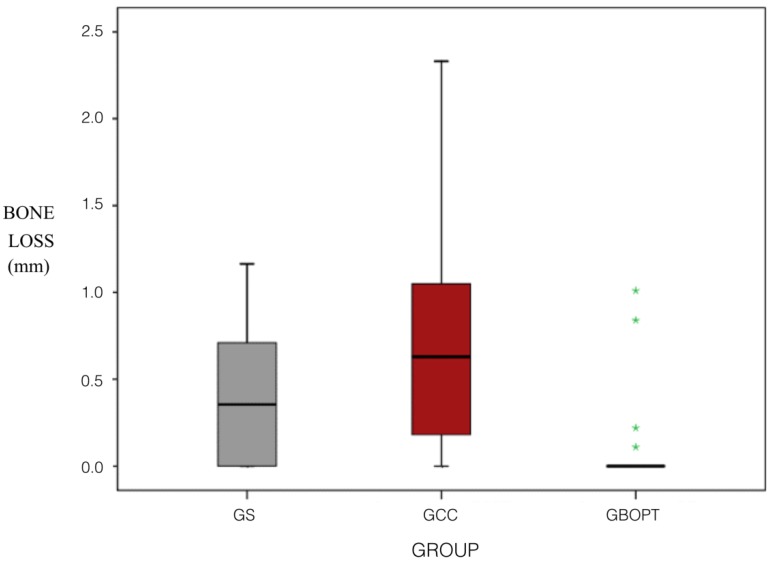
Box plot showing bone loss (mm) according to prosthetic type. GS, screw-retained crown; GBOPT, cemented without finishing line; GCC, conventional cementation with finishing line. The asterisks indicate dispersed values in statistical analysis. In this case, the dispersed values are given in the BOPT group.

**Figure 7 jcm-08-02183-f007:**
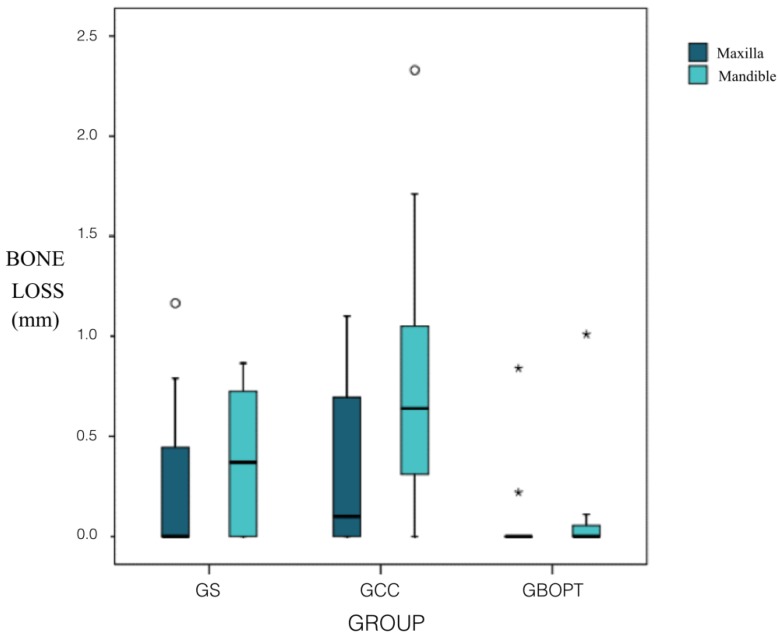
Box plot showing bone loss (mm) in relation to implant position (maxillary/mandibular). The asterisks and circles indicate dispersed values in statistical analysis.

**Figure 8 jcm-08-02183-f008:**
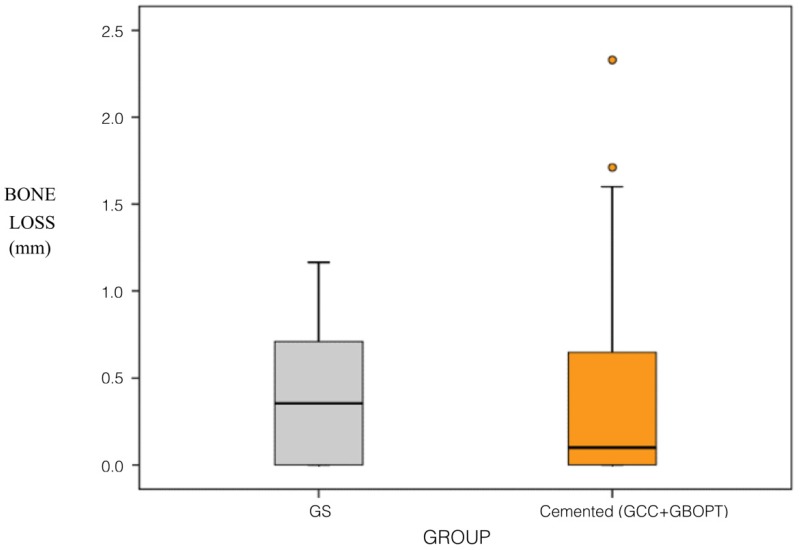
Box plot showing bone loss (mm) in relation to prosthetic fixation (screwed versus cemented). The circles indicate dispersed values in statistical analysis.

**Table 1 jcm-08-02183-t001:** Demographic data and clinical characteristics.

	Female	Male	Total
Number of Patients	49	26	75
Number of Implants	49	26	75
Mean Age	−	−	42.7 ± 10.6

**Table 2 jcm-08-02183-t002:** Bone loss (mm) in relation to prosthetic type.

		GROUP		
	TOTAL	Screw-Retained	Cemented BOPT	Cemented Conventional
N	68	25	21	22
Mean	0.38	0.35	0.10	0.67
Standard deviation	0.50	0.37	0.28	0.62
Minimum	0.00	0.00	0.00	0.00
Maximum	2.33	1.17	1.01	2.33
Median	0.14	0.36	0.00	0.63

**Table 3 jcm-08-02183-t003:** Simple linear regression model with the variables analyzed.

Model	Unstandardized Coefficients		Standardized Coefficients	*t*	Sig.	95% Confidence Interval for B	
	B	Std. Error	Beta			Lower Bound	Upper Bound
(Constant)	0.173	0.273		0.634	0.529	−0.373	0.719
GCC	0.303	0.128	0.288	2.364	0.021	0.047	0.560
GBOPT	−0.160	0.132	−0.150	−1.213	0.230	−0.423	0.104
ARCH (mdb)	0.047	0.131	0.047	0.356	0.723	−0.215	0.308
MOLAR (yes)	0.273	0.129	0.276	2.117	0.038	0.015	0.531
SEX (female)	−0.097	0.111	−0.097	−0.879	0.383	−0.319	0.124
AGE	0.001	0.005	0.016	0.149	0.882	−0.010	0.011

“B”: unstandardized coefficient; “Std. Error”: standard error; “*t*”: t-student statistic value for the nullity test of the coefficient; “Sig.”: significance probability.

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
