# Peer review of "Influence of Biologically Oriented Preparation Technique on Peri-Implant Tissues; Prospective Randomized Clinical Trial with Three-Year Follow-Up. Part I: Hard Tissues"

_jcm, 2019, doi:10.3390/jcm8122183_

Round 1
Reviewer 1 Report
1. Sample size calculation not reported.
2. It would have been interesting to report also the bleeding on probing data, associating it to both bone loss and the type of prosthesis.
Author Response
Manuscript ID: jcm-651658
Title: Influence of biologically oriented preparation technique on
peri-implant tissues; prospective randomized clinical trial with three-year
follow-up. Part I: hard tissues.
-Reviewer 1 –
Comments and Suggestions for Authors
Sample size calculation not reported.
Reply: Thank you for your comment. This is the explanation of how the sample calculation was performed. It has been introduced in the manuscript.
2.4. Sample size calculation
A previous study of the sample size (power analysis) was carried out, concluding that a minimum of 75 implants (25 per group) were needed to achieve a power of 80% in order to detect as significant an effect size f= 0.35 (half-large) in the average MBL of groups, for 95% confidence. Standard deviation 0.45mm was assumed, so that effect size is compatible with average MBL in groups 0.25mm, 0.45mm and 0.65mm.
It would have been interesting to report also the bleeding on probing data, associating it to both bone loss and the type of prosthesis.
Reply: Thank you for your comment. Since this work is part of extensive research that also analyses the behavior of soft tissues, the bleeding on probing data is recorded in the article about the behavior of soft tissues that is in the same journal in the way of publication. This is a brief summary of the results:
One of the variables analyzed was the presence of BOP. In Group GS (screw-retained), 24.0% of the sample presented BOP. In Group GBOPT (cemented BOPT prostheses), 9.5% of the sample presented BOP, while in Group GCC (conventional cemented prostheses) 40.9% of cases presented BOP. Differences in BOP rates were situated on the threshold of significance (P=0.059). Differences were clearly significant between Groups GBOPT and GCC (P=0.018, Chi2 test) but not between Groups GS and GCC (P=0.215, Chi2 test), or between Groups GS and GBOPT (P=0.260, Chi2 test).

Reviewer 2 Report
This paper sets out to analyse and compare the clinical behaviour of three types of prosthesis supported by one-piece implants in the posterior region after three years functional loading. Adopting a randomised clinical trial methodology, the authors conclude that bone loss around implants is related to the type of prosthesis restoration that the implant supports and that the sub-group defined as those in which a ‘biologically orientated preparation technique’ (GBOPT) suffered the least bone loss over the study period.
The paper is well written and the authors have recognised the limitations of their work due to the small numbers involved in the study. The work in my view could be improved through a more detailed presentation of the findings from individual groups. There would appear to be considerable variation in the GBOPT group, indeed I struggle to understand the graphics used in Figure 5 and 6, especially the former, in which bone loss for 1 individual is double the maximum of any of the other sub groups although the chart does not show the standard deviation.
With respect to analyses I wonder whether the authors could consider some form of regression analyses to explore sex and age as factors as well as position. Indeed, I would take issue with the comment on lines 194 and 195, ‘..fell slightly short..). When undertaking a statistic test it either is significant or it is not and I would suggest the wording is changed.
Finally, I would have liked to have seen some discussion on the implications for clinicians. Do the authors feel that in clinical terms the differences between the groups are of importance?
Author Response
Manuscript ID: jcm-651658
Title: Influence of biologically oriented preparation technique on
peri-implant tissues; prospective randomized clinical trial with three-year
follow-up. Part I: hard tissues.
-Reviewer 2 –
Comments and Suggestions for Authors
This paper sets out to analyse and compare the clinical behaviour of three types of prosthesis supported by one-piece implants in the posterior region after three years functional loading. Adopting a randomised clinical trial methodology, the authors conclude that bone loss around implants is related to the type of prosthesis restoration that the implant supports and that the sub-group defined as those in which a ‘biologically orientated preparation technique’ (GBOPT) suffered the least bone loss over the study period.
The paper is well written and the authors have recognised the limitations of their work due to the small numbers involved in the study. The work in my view could be improved through a more detailed presentation of the findings from individual groups.
Reply: Thank you for your comment. The text has been revised introducing the explanations as you advise.
There would appear to be considerable variation in the GBOPT group, indeed I struggle to understand the graphics used in Figure 5 and 6, especially the former, in which bone loss for 1 individual is double the maximum of any of the other sub groups although the chart does not show the standard deviation.
Reply: Thank you for your comment the article has been modified accordingly because it was a mistake. The bone loss of the three restoration types as well as the SD are explained bellow. They were also on the table 2, but the groups BOPT and cemented convencional were wrong by an error during the translation process. Sorry for the inconvenience.
Radiographic peri-implant bone loss showed differences between the three types of restoration used: Group GS (screw-retained) = 0.35mm ± 0.37mm; Group GBOPT (cemented BOPT) = 0.10mm ± 0.28mm; and Group GCC (conventional cementation with finishing line) = 0.67mm ± 0.62 mm. Median bone loss values were: 0.36mm; 0.63mm; and 0.00 mm, respectively (Table 2).
Table 2. Bone loss (mm) in relation to prosthetic type.
|
|
|
GROUP |
|
|
TOTAL |
Screw-retained |
Cemented BOPT conventional |
Cemented BOPT conventional |
N |
68 |
25 |
22 |
21 |
Mean |
0.38 |
0.35 |
0.67 |
0.10 |
Standard deviation |
0.50 |
0.37 |
0.62 |
0.28 |
Minimum |
0.00 |
0.00 |
0.00 |
0.00 |
Maximum |
2.33 |
1.17 |
2.33 |
1.01 |
Median |
0.14 |
0.36 |
0.63 |
0.00 |
Bone loss (mm) for the three types of prosthesis and two cementation techniques (BOPT/conventional) was analyzed. No statistically significant differences were found between GS and GCC (P = 0.059, Kruskal-Wallis and Mann-Whitney tests), although less bone loss was observed in Group GS. However, significant differences were found between Groups GS and GBOPT (P=0.013, Kruskal-Wallis and Mann-Whitney tests) whereby the GBOPT Group underwent less bone loss. In the same way, differences were also found between the GBOPT and GCC Groups (P = <0.001, Kruskal-Wallis and Mann-Whitney tests), Group GBOPT presenting less bone loss. The rate of incidence of bone loss for the three prosthetic types and two cementation techniques (BOPT/conventional) showed differences between Group GS and Group GBOPT (P= 0.011, Chi2) and between Group GBOPT and Group GCC (P=<0.001, Chi2) although the difference but not between Group GS and Group GCC fell slightly short of statistical significance (P=0.058, Chi2), although Group GS presented presenting less bone loss than GCC Group in Group GBOPT (Figure 5).
With respect to analyses I wonder whether the authors could consider some form of regression analyses to explore sex and age as factors as well as position.
Reply: Thank you for your comment. The explanation has been introduced as you advise. I hope it would be clarifying. It has been introduced in the manuscript as follows:
3.3. Regression analysis to explore sex, age and position factors
A regression model to explain the MBL based on gender and age variables, as well as the position of the implant was made. The intention was to assess its direct influence on MBL and adjust the main relationship (the effect of the type of restoration) by this series of variables that may exert some kind of confusion in the event that the groups of restorations are not homogeneous and there is some kind of bias. The model-dependent variable was the MBL. Independent variables were: Group (screwed/cemented conventional /cemented BOPT), Location (maxilla/mandible), Position (premolar/molar) , Sex (male/female) and Age. The analyses result in fairly homogeneous distributions in the 3 groups of restorations. Note that the overall average age fits perfectly with the data registered: 43.4 ± 10.4 years. The interpretation of the model and its coefficients is represented as follows (Table 3):
Table 3. Simple linear regression model with the variables analyzed. |
||||||||
Model |
Unstandarized Coefficients |
Standarized Coefficients |
t |
Sig. |
95% Confidence interval for B |
|||
B |
Std. Error |
Beta |
Lower Bound |
Upper Bound |
||||
|
(Constant) |
0.173 |
0.273 |
|
0.634 |
0.529 |
-0.373 |
0.719 |
GCC |
0.303 |
0.128 |
0.288 |
2.364 |
0.021 |
0.047 |
0.560 |
|
GBOPT |
-0.160 |
0.132 |
-0.150 |
-1.213 |
0.230 |
-0.423 |
0.104 |
|
ARCH (mdb) |
0.047 |
0.131 |
0.047 |
0.356 |
0.723 |
-0.215 |
0.308 |
|
MOLAR (yes) |
0.273 |
0.129 |
0.276 |
2.117 |
0.038 |
0.015 |
0.531 |
|
SEX (female) |
-0.097 |
0.111 |
-0.097 |
-0.879 |
0.383 |
-0.319 |
0.124 |
|
AGE |
0.001 |
0.005 |
0.016 |
0.149 |
0.882 |
-0.010 |
0.011 |
An important factor in explaining the MBL is the position. In the molars there is a MBL significantly higher than that of the premolars (beta= 0.273; P= 0.038). The maxilar/mandible position or demographic profile has no relevant influence on the MBL. Regarding the quality of fit of the model, the coefficient of determination R2= 31% and the adjusted R2 adjusted= 24.2%, which means that beyond the findings in terms of significant relationships, the equation is not good for making accurate MBL predictions. Despite the lack of normalcy of the dependent variable, especially in some groups with the cemented BOPT, the model residues fit reasonably well to the normal pattern, even with some tendency to positive asymmetry.
Indeed, I would take issue with the comment on lines 194 and 195, ‘..fell slightly short..). When undertaking a statistic test it either is significant or it is not and I would suggest the wording is changed.
Reply: Thank you for your comment and sorry for the inconvenience because again it was a mistake on the last sentence, the comparation was between GS and GCC group (not GBOPT) , which not presented differences although bone loss was less in GS Group.
In the same way, differences were also found between the GBOPT and GCC Groups (P = <0.001, Kruskal-Wallis and Mann-Whitney tests), Group GBOPT presenting less bone loss. The rate of incidence of bone loss for the three prosthetic types and two cementation techniques (BOPT/conventional) showed differences between Group GS and Group GBOPT (P= 0.011, Chi2) and between Group GBOPT and Group GCC (P=<0.001, Chi2) although the difference but not between Group GS and Group GCC fell slightly short of statistical significance (P=0.058, Chi2), although Group GS presented presenting less bone loss than GCC Group in Group GBOPT (Figure 5).
Finally, I would have liked to have seen some discussion on the implications for clinicians. Do the authors feel that in clinical terms the differences between the groups are of importance?
Reply: Thank you for your comment the text has been revised introducing the implications as you advise.
This prospective RCT could help on clinicians election for the restoration of an implanted-supported unit posterior region because, in terms of bone loss, it would be recommended to place BOPT cemented restoration, or screwed CAD-CAM restoration direct to implant because if it is not not considered the improvement in bone loss that it is produced by cemented crown without finishing line, screw-retained implant crowns are more favoured by the clinician, due to their reduced risk of biological complications as a consequence of remaining excess cement and its reversibility in case of complications related to the need for removal of the restoration.

Reviewer 3 Report
This article analyzed and compared the clinical behavior of three types of prosthesis supported by implants in the posterior region after three years of functional loading.
The article is well written. The introduction, M&M section and results section are good but can be improved a bit.
There is one sentence in chapter 2.5 which is totally confusing. Did the authors really only take the radiographs from prosthetic loading to 3(!) months after loading into account for the measurements in an official 3 years(!) follow-up study? That would be totally insufficient! The main bone remodelling takes place during the first 6 months up to first year! Please state clearly your procedures including the time intervals of the radiographs that you used for the examination.
The discussion section is too short and must be improved. Please follow the STROBE guidelines. Please discuss your results and not only repeat them and cite studies.
After a revision, the article may be acceptable.

Author Response
Manuscript ID: jcm-651658
Title: Influence of biologically oriented preparation technique on
peri-implant tissues; prospective randomized clinical trial with three-year
follow-up. Part I: hard tissues.
-Reviewer 3 –
Comments and Suggestions for Authors
This article analyzed and compared the clinical behavior of three types of prosthesis supported by implants in the posterior region after three years of functional loading.
The article is well written. The introduction, M&M section and results section are good but can be improved a bit.
Reply: Thank you for your comment the text has been revised introducing the improvements as you advise.
There is one sentence in chapter 2.5 which is totally confusing. Did the authors really only take the radiographs from prosthetic loading to 3(!) months after loading into account for the measurements in an official 3 years(!) follow-up study? That would be totally insufficient! The main bone remodelling takes place during the first 6 months up to first year!Please state clearly your procedures including the time intervals of the radiographs that you used for the examination.
Reply: Thank you for your comment the text has been revised and it was a mistake. It was wrong by an error during the translation. The measurements were taken at the moment of prosthetic loading (T0) and again 3 years after loading. Although follow-up radiographs were performed because patients returned for follow-up check-ups after 1 month, 6 months, and 1, 2, and 3 years, the standardized ones for the study were performed at the time of loading and 3 years later.
Measurements were taken to evaluate bone changes between T0 (prosthetic loading) and T1 (3 months years later).
The discussion section is too short and must be improved. Please follow the STROBE guidelines. Please discuss your results and not only repeat them and cite studies.
Reply: Thank you for your comment the text has been revised introducing the discussion improvement as you advise.

Round 2
Reviewer 3 Report
Dear authors,
thank you for implementation of my recommendations.
The quality of your paper has significantly increased.
I recommend to accept it in the present form.
Kind regards